# The Regulation of m6A Modification in Glioblastoma: Functional Mechanisms and Therapeutic Approaches

**DOI:** 10.3390/cancers15133307

**Published:** 2023-06-23

**Authors:** Simon Deacon, Lauryn Walker, Masar Radhi, Stuart Smith

**Affiliations:** 1Children’s Brain Tumour Research Centre, University of Nottingham, Nottingham NG7 2RD, UK; 2Nottingham University Hospitals NHS Trust, Nottingham NG7 2UH, UK

**Keywords:** glioblastoma, epitranscriptomics, RNA methylation

## Abstract

**Simple Summary:**

Brain tumours are a leading cause of cancer-related death in people under the age of 40, but most remain untreatable. There is an urgent need for a greater understanding of tumour biology to help guide the development of new treatments. We hypothesise that a type of genetic code—RNA—may be key to understanding how these tumours develop, and so how to treat them. Evidence has shown that the modification of this RNA is altered in brain tumours, and this may serve to regulate diverse mechanisms including tumour proliferation, invasion and treatment evasion. However, the precise mechanisms by which these RNA modifications exert their functional effects are poorly understood. This review summarises the evidence for this disordered regulation of RNA methylation in glioblastoma, and explores the downstream functional effects of such modification on RNA fate.

**Abstract:**

Glioblastoma is the most prevalent primary brain tumour and invariably confers a poor prognosis. The immense intra-tumoral heterogeneity of glioblastoma and its ability to rapidly develop treatment resistance are key barriers to successful therapy. As such, there is an urgent need for the greater understanding of the tumour biology in order to guide the development of novel therapeutics in this field. N6-methyladenosine (m6A) is the most abundant of the RNA modifications in eukaryotes. Studies have demonstrated that the regulation of this RNA modification is altered in glioblastoma and may serve to regulate diverse mechanisms including glioma stem-cell self-renewal, tumorigenesis, invasion and treatment evasion. However, the precise mechanisms by which m6A modifications exert their functional effects are poorly understood. This review summarises the evidence for the disordered regulation of m6A in glioblastoma and discusses the downstream functional effects of m6A modification on RNA fate. The wide-ranging biological consequences of m6A modification raises the hope that novel cancer therapies can be targeted against this mechanism.

## 1. Introduction

The epitranscriptional methylation of RNA at the internal N6-adenosine (m6A) is the most common internal base post-transcriptional modification of messenger RNAs, with more than a quarter of mammalian transcripts carrying these marks [1]. More recent m6A mapping has demonstrated that its distribution is preferentially located near stop codons and at the 3′ untranslated region (3′ UTRs) [2,3]. However, recent single-cell m6A-sequencing has demonstrated a high degree of m6A heterogeneity between single cells and across the cell cycle [4]. Physiological functions in which m6A have been implicated are many and varied, with the m6A modification now established as a key determinant of RNA fate and metabolism [5]. Proteins involved in the regulation of m6A methylation have been characterised into three functional groups: “writers”, “erasers” and “readers”, which, respectively, methylate, demethylate and recognise the m6A modification (Figure 1).

### 1.1. Writers

Two subcomplexes together form the methyltransferase complex, catalysing the addition of m6A to mRNA at a conserved m6A consensus site RRACH. This group consists of a canonical m6A-METTL Complex (MAC), formed of the METTL3-METTL14 heterodimer, and the additional m6A-METTL Associated Complex, (MACOM), composed of WTAP, RBM15, VIRMA, ZC3H13 and HAKAI [6]. This METTL3–METTL14–WTAP methyltransferase complex is predominantly active within the nuclear speckles, mediating the co-transcriptional deposition of m6A in nascent transcripts [7,8]. The MACOM complex regulates the writer complex cellular location and binding specificity, with evidence that VIRMA directs activity of the complex to the STOP codon and 3′UTR of mRNA [9], and WTAP and VIRMA directing the nuclear localisation of the complex [10,11].

### 1.2. Erasers

Thus far, two proteins have been demonstrated to have demethylase activity, catalysing the removal of m6A from mRNA transcripts. Fat mass and obesity-associated protein (FTO) was the first demethylase to be discovered to oxidatively reverse m6A to adenosine [12]. The second demethylase protein to be identified was α-ketoglutarate-dependent dioxygenase homolog 5 (ALKBH5) [13]. The localisation of FTO has been shown to be context-dependant and variable between cells, with differential localisation of FTO in both the nucleus and cytoplasm between cell lines [14]. Crucially, the discovery of a class of demethylases has raised the notion of m6A being a dynamic modulation, potentially adaptive to cellular stimuli and stress [5]. However, Bulk MeRIP–Seq analysis of FTO-knockout brain tissue showed that m6A levels were largely unaffected, apart from in a subset of mRNAs where m6A peaks were higher, [15] and it has been argued that FTO has a much higher affinity for demethylating another epitranscriptomic modification, m6Am, rather than m6A [16]. Furthermore, ALKBH5-knockout mice appear normal except for defective spermatogenesis, with ALKBH5 enriched in reproductive organs [13], suggesting a limited systemic role of this gene in normal physiological conditions. Furthermore, the apparent differences in m6A levels between experimental conditions observed in some studies may be explained by limitations of the sequencing methodology used [17]. As such, recent work has questioned the notion of the dynamic regulation of m6A modification [8,18], and further research is required to solve this puzzle.

### 1.3. Readers

“Readers” are a diverse category of proteins which contain specific domains enabling them to recognise and bind to m6A-modified RNA at the consensus site. They exert differential downstream effects on the mRNA strands, including splicing, nuclear export, degradation, stabilisation and translation [19]. The most robustly classified family of proteins within this category are those containing the YT521-B homology (YTH) domain: YTH Domain Family 1-3 (YTHDF1-3) and YTH Domain Containing 1-2 (YTHDC1-2). Some studies have demonstrated opposing functions for these proteins. Cytoplasmic YTHDF2 promotes the degradation of target transcripts, whereas YTHDF1 and YTHDF3 promote target translation [20,21,22,23,24]. On the other hand, other authors have argued that the YTHDF family represent redundant proteins which share the dual functions of RNA degradation and translation promotion [25,26]. The YTHDF proteins contain a large low-complexity region which is understood to undergo phase separation within the cytoplasm, drawing mRNA transcripts to membrane-less bodies such as P-bodies and stress granules [27]. Whilst the activity of the YTHDF-family is cytoplasmic, YTHDC1 has been demonstrated to have localisation within nuclear speckles, with functions including mediating mRNA splicing and epigenetic silencing by the ncRNA *XIST* [28,29].

Other known m6a readers include heterogeneous nuclear ribonucleoprotein C (HNRNPC), heterogeneous nuclear ribonucleoprotein G (HNRNPG), and A2B1 (HNRNPA2/B1) [30,31]. These belong to a class of indirect readers which, unlike proteins containing the YTH-domain, bind to m6a-labelled mRNA as a consequence of structural changes induced by m6a, making their RNA-binding site more accessible, and thus altering binding efficiency [31]. Another family of RNA-binding proteins, IGF2BP1-3, have been demonstrated to enhance stability of m6A-modified mRNAs [32]. It is not yet clear whether IGF2BP proteins bind to the m6A site directly, bind due to m6A-mediated changes in mRNA structure, or bind to other m6A readers at the site [29,33].

## 2. m6A Regulation in Glioblastoma

Previous studies have demonstrated the crucial role of m6A in nervous-system development [34,35]. FTO is highly expressed in the brain and has been shown to regulate neurogenesis during development [15,36]. Knockout of the m6A reader YTHDF2 demonstrated lethality in mice, with severely compromised neural development and compromised neural stem/progenitor cell self-renewal [37]. Similarly, knock-out of *METTL14* in embryonic neural stem cells markedly reduced their proliferation and differentiation [38]. Intriguingly, the role of m6A in stem-cell self-renewal and differentiation leads to the hypothesis that this process may be disordered in many different types of cancer [39,40]. Glioblastoma is hypothesised to arise from stem-like precursor cells which dynamically initiate and maintain the diverse tumour cell population [41,42]. The heterogeneity of GBM contributes to its ability to plastically adapt to environmental conditions and evade treatment [43]. As such, the dynamic m6A modification may be a mechanism by which glioma stem cells adapt to environmental conditions and develop treatment resistance [44,45]. Indeed, studies have shown that expression patterns of these regulatory genes correlate with glioma grade and demonstrate potential use in prognostic stratification [46,47,48].

### 2.1. Writers

Multiple papers have reported that the m6A writer METTL3 is upregulated in glioma stem cells (GSCs) and contributes to tumorigenesis [49,50,51,52,53,54]. METTL3 silencing reduced the expression of the glioma reprogramming factors POU3F2, SOX2, SALL2 and OLIG2, and suppressed GSC proliferation in neurosphere assays derived from human glioma cell lines. The authors also highlighted a direct role for METTL3 in mediating radioresistance, showing that METTL3-knockdown GSCs were more sensitive to radiotherapy. The mRNA transcript targets of METTL3 with the highest binding affinity were the reprogramming factors *SOX2* and *SOX21*. Furthermore, METTL3 silencing leads to a reduction in *SOX2* mRNA levels, and in vivo studies found a significant positive correlation at the protein level between SOX2 and METTL3 expression in GBM tissue samples. Mechanistically, it was found that METTL3-mediated, m6A modification increased the stability of *SOX2* mRNA. *SOX2* has previously been shown to have a role as a reprogramming factor in GSCs [55]. *HuR* transcript levels were also increased in both GSCs and GBM brain samples compared with controls, and silencing of *HuR* in GSCs led to the failure of the METTL3-mediated stabilisation of *SOX2* transcripts. Previous studies have illustrated the independent involvement of *HuR* in glioma tumour growth and chemoresistance [56]. Interestingly, subsequent research has demonstrated that HuR is necessary for the METTL3-mediated stabilisation of metastasis-associated lung adenocarcinoma transcript 1 (MALAT1), which in turn activates nuclear factor kappa B (NFB) in isocitrate dehydrogenase (IDH) wildtype glioma [50]. The same study found that METTL3 expression is positively associated with a higher grade and poorer prognosis in IDH-wildtype glioma, but not in IDH-mutant gliomas.

Whole-transcriptome sequencing and m6A-RNA-immunoprecipitation-coupled sequencing (m6A-RIP Seq) has been utilised to explore the mRNA targeted by METTL3, by identifying differentially expressed and m6A-modified transcripts in GSCs with non-targeting shRNA or METTL3 shRNA silencing [51]. The study corroborated the findings that m6a modifications are concentrated in the 3′UTR region and that METTL3 depletion suppresses both m6A abundance and GSC proliferation. Transcripts containing a METTL3-dependent m6A peak were downregulated at the transcript level in METTL3-silenced GSCs, suggesting that m6A modification mediates the RNA stabilisation of these transcripts. Gene Set Enrichment Analysis (GSEA) revealed a significant depletion of genes implicated in oncogenic pathways following METTL3-silencing, including MYC, mTORC1, E2F, TG-F/NFKB, and cell-cycle and DNA-repair pathways. This supports the notion that METTL3-mediated m6A modification sustains the high expression of genes that are essential for GSC maintenance and tumorigenesis. The m6A modification may be adaptive to cellular stressors such as therapeutics, and studies have shown that m6A modification is significantly increased in temozolomide-resistant GBM cells and that temozolomide treatment induces the upregulation of *METTL3*, thus leading to increased m6a methylation [52,57].

A novel mechanism for the oncogenic function of METTL3 in GBM is by mediating nonsense-mediated mRNA decay (NMD) [53]. The authors corroborated previous findings by showing that METTL3 levels are upregulated in GBM patient samples and that, in mice models, METTL3-knockdown suppressed tumour growth. Additionally, RNA sequencing found that carcinogenesis-associated pathways, including apoptotic signalling pathways, were enriched in the set of m6A-regulated genes in *METTL3*-silenced GBM lines. METTL3 knockdown in GBM cells increased the activation of NMD which in turn increased the degradation of splicing factor transcripts. This process was mediated by the nuclear m6A reader YTHDC1. The METTL3-mediated disruption of alternative splicing was found to induce a neomorphic function in genes implicated in tumorigenesis. Expression of the pro-apoptotic isoform of BCL-X, BCL-XS, was increased in METTL3-knockdown cells, whereas the anti-apoptotic isoform BCL-XL was increased in wild-type GBM cells.

In contrast with the findings discussed thus far, two studies have provided evidence suggesting that METTL3, and the associated increase in m6a levels, may have a tumour-suppressive role in GBM. Both METTL3 and METTL14 knockdown reduced m6A levels and increased the self-renewal of GSCs, indicating that upregulated levels of METTL3 may in fact suppress GSC maintenance. METTL3 and METTL14 knockdown GSCs in a mouse model of GBM produced larger tumours with poorer survival rates [58]. Oncogenes including ADAM19 and EPHA3 were upregulated and tumour suppressor genes were downregulated in METTL3/14-knockdown cells, compared to controls. Consistent with these findings, a recent paper demonstrated that m6A levels were reduced in GBM cells and that decreased methylation corresponded with an increased epithelial-to-mesenchymal transition and vasculogenic mimicry [59]. Notably, both studies also demonstrated that increased expression of the m6A erasers *FTO* or *ALKBH5* had tumorigenic effects. The different effects ascribed to *METTL3* and m6A methylation in these studies are possibly explained by the intrinsic genetic heterogeneity of GBM, which could account for context-dependent variation in the downstream effects of m6A modification on cellular function. However, this will undoubtedly pose a challenge when developing therapeutics targeting m6A methylation and its regulators.

The m6A modification has been implicated in RNA editing in glioblastoma, with METTL3 altering A-to-I and C-to-U events via the differential regulation of RNA editing enzymes ADAR and APOBEC3A [51]. A more recent study has shown, using both glioma cell lines and mouse models, that METTL3 methylates ADAR1 mRNA which, mediated by YTHDF1, led to its increased expression at the protein level [54]. ADAR1 promoted tumorigenesis by acting as an RNA-binding protein, binding and stabilising the cyclin-dependent kinase 2 (CDK2) mRNA transcript, a crucial cell-cycle kinase.

### 2.2. Erasers

The overexpression of ALKBH5, and thus the subsequent decrease in m6a levels, has been shown to contribute to the maintenance and proliferation of GSCs [60]. Analysis of the TCGA found that there was a significant negative correlation between an increase in ALKBH5 levels and patient prognosis, suggesting that ALKBH5 could be a prognostic marker in GBM. Moreover, in vitro experiments showed that ALKBH5 expression was upregulated in patient-derived GSCs. ALKBH5 knockdown in GSCs significantly reduced proliferation and tumour sphere formation in GSC cell lines. Furthermore, ALKBH5-knockdown GSC cells in a mouse model demonstrated lower rates of tumour formation and an improved prognosis. The transcription factor FOXM1 was noted as a downstream target gene of ALKBH5, consistent with previous studies which found FOXM1 to have a critical role in the tumorigenicity of GSCs, in addition to being associated with poor prognosis in GBM patients [61,62]. ALKBH5 knockdown increased m6A levels of FOXM1 pre-mRNA, which subsequently decreased the expression of FOXM1. ALKBH5-mediated demethylation of *FOXM1* promoted the recruitment of the RNA-binding protein HuR to *FOXM1*, increasing its expression. This is supported by TCGA data showing a significant positive correlation between ALKBH5 and FOXM1 mRNA expression. Therefore, it can be inferred that ALKBH5-mediated m6a demethylation indirectly mediates the maintenance and proliferation of GSCs by upregulating the expression of FOXM1 which subsequently exerts its downstream effects in terms of promoting GSC maintenance and proliferation.

The rapid growth of glioblastoma causes significant changes to the tumour microenvironment which the tumour must rapidly adapt to, including significant hypoxic stress. Dynamic m6A modification is a putative mechanism in this process, and a study in GBM cells has shown the hypoxia-induced upregulation of ALKBH5 in a dependant manner in glioblastoma-derived cell culture [63]. Transcripts m6a-demethylated by ALKBH5 were demonstrated in a ALKBH5-knockdown condition, with the lncRNA NEAT1 a significant target for demethylation. NEAT1 is necessary for paraspeckle assembly, which mediates a range of cellular processes such as transcription and splicing. ALKBH5 facilitates paraspeckle assembly through m6A demethylation and the stabilization of NEAT1. A further study has demonstrated hypoxia-induced ALKBH5 upregulation in breast cancer, with the knockdown of ALKBH5 reducing tumour initiation, potentially demonstrating a conserved mechanism by which m6A contributes to tumour microenvironmental adaptation [64].

Beyond adaptation to the tumour environment, another significant plastic change in tumours is in the development of treatment resistance. A study has demonstrated the role of ALKBH5 in promoting radio-resistance in GBM through mediating homologous repair mechanisms in patient-derived glioblastoma cell culture [65]. In cells downregulated for ALKBH5, there was decreased survival of GSCs after irradiation, with a concomitant decreased expression of genes involved in homologous repair, including CHK1 and RAD51. Furthermore, ALKBH5 was demonstrated to contribute to the invasiveness of GBM, with GSCs deficient for ALKBH5 showing significantly reduced invasion capability.

FTO was associated with GBM when it was reported that the FDA-approved FTO-inhibitor meclofenamic acid (MA2) suppressed the proliferation of GSCs across five different glioblastoma-derived cell lines, and that MA2 administration in mice suppressed xenograft GBM tumour growth. These findings suggested that FTO may have a role in GSC maintenance and in turn, GBM tumorigenesis [58]. A possible mechanism for this effect is the role FTO has in the context of the frequent IDH1/2 mutations in gliomas. IDH1 and IDH2 mutations have long been established in the molecular pathology of gliomas, and now form a core aspect of their diagnostic classification [66]. These enzymes catalyse the oxidative decarboxylation of isocitrate to α-ketoglutarate (α-KG). IDH mutations alter enzyme activity whereby α-KG is converted to the oncometabolite R-hydroxyglutarate (R-2HG). R-2HG is a competitive inhibitor of both FTO and ALKBH5. Hence, it was hypothesised that mutant IDH1/2 isoforms may indirectly increase m6A levels by the competitive inhibition of the demethylases FTO and ALKBH5 [67].

HEK293T cells expressing mutant IDH2 had much higher levels of m6A RNA than the control, and exposing cells to an α-KG analogue together with a competitive inhibitor significantly increased m6a levels, confirming R-2HG to be a competitive inhibitor of the m6a demethylases [67]. In vivo experiments in a mouse model of AML showed that R-2HG significantly inhibited AML progression and prolonged survival in mice who were xenografted with IDH wild type R-2HG-sensitive cells [68]. R-2HG produced endogenously by IDH1/2 mutant cells delayed AML progression. These findings indicate the tumour-suppressive role of R2-HG. When IDH1/2 mutant leukaemia cell lines were exposed to R-2HG, this anti-proliferative activity was not observed, suggesting that the presence of the IDH1/2 mutation confers tolerance to R-2HG. RNA sequencing showed that FTO was expressed at a significantly higher level in R-2HG-sensitive cells compared to -resistant cells. R-2HG treatment was found to increase m6A abundance in sensitive cells by inhibiting FTO demethylase activity. Increased m6A lead to the decreased transcript stability, and expression, of the oncogenes *MYC* and *CEBPA*. This, therefore, suggests that high levels of FTO maintains the self-renewal ability of tumour cells through the stabilisation of oncogenes, consistent with other studies [69,70]. In glioma cell lines, R-2HG inhibited their proliferation in IDH1/2 wild-type lines. IDH1/2 mutations are predominantly associated with lower-grade glioma, and it may be the case that endogenous R-2HG production in GSCs suppresses FTO levels, thus reducing oncogene expression and limiting progression by maintaining m6a levels. FTO inhibition has been shown to enhance the effect of temozolomide, via targeting the MYC-miR-155/23a Cluster-MXI1 feedback circuit [71].

### 2.3. Readers

The downstream effects of m6A modification are dictated by the binding of reader proteins to their target RNAs, and these proteins are thus essential for dictating RNA fate. The YTHDF protein family are the canonical cytoplasmic mRNA m6A readers and are associated with glioma progression. YTHDF1 has been shown to predict poor prognosis in glioma patients, and the knockdown of YTHDF1 in GBM inhibits proliferation and reduces resistance to temozolomide in both patient-derived cell lines and xenograft models [72,73]. Furthermore, Musashi-1, an RNA-binding protein which post-transcriptionally regulates gene expression, was found to upregulate YTHDF1 protein expression via stabilising YTHDF1 mRNA [73].

YTHDF2 has been shown to contribute to glioblastoma cell proliferation and tumorigenesis, and overexpression was mediated in glioblastoma via the EGFR/SRC/ERK pathway [74]. YTHDF2 in glioblastoma mediated the degradation of UBX domain protein 1, in turn inducing NF-KB activation [75]. A more recent study has shown that YTHDF2 contributed to poor prognosis and was upregulated in temozolomide-resistant cell lines [76]. Its effect was mediated by the degradation of EPHB3 and TNFAIP3 mRNA which also led to the induction of NF-KB. The YTHDF family of proteins have redundant functions in both mRNA degradation and stabilisation [26]. In this vein, a separate study found that YTHDF2 mediates the stability of MYC and VEGFA mRNA in GSCs [77]. YTHDF3 has been demonstrated to promote breast-cancer brain metastasis, although no studies have demonstrated a role in glioma pathology [21,78].

Interestingly, YTHDF1 may exert a tumorigenic effect via its modulation of the inflammatory microenvironment. YTHDF1-deficient mice demonstrated an increased antigen-specific CD8+ T cell antitumour response in a rodent model of B16 melanoma [79]. Similarly, the expression of YTHDF2 has been shown to correlate with immune-cell and tumour-associated-macrophage markers and was associated with poor survival in low-grade glioma [80].

YTHDC1 has been demonstrated to contribute to tumorigenesis, with deficient cell lines demonstrating impaired sphere formation [53]. YTHDC1 binding led to the nonsense-mediated mRNA decay of Serine/arginine-rich splicing factor (SRSF) transcripts in a m6A-dependent manner. Conversely, another study demonstrated that YTHDC1 impaired proliferation by reducing the expression of VPS25 [81]. VPS25 is highly expressed in glioma and increases proliferation via JAK-STAT signalling. Intriguingly, SRSF7 has been demonstrated to be a novel m6A regulator, which exerts a tumorigenic effect in glioblastoma via its modulation of m6A sites on PDZ-binding kinase, and via binding by the m6A reader IGF2BP2 [82].

Heterogeneous nuclear ribonucleoprotein C (HNRNPC) and heterogeneous nuclear ribonucleoprotein A2/B1 (HNRNPA2/B1) are proteins within the family of indirect readers. HNRNPC and HNRNPA2/B1 are highly expressed in glioblastoma cell lines and brain tissue [83], and the increased expression of HNRNPA2/B1 protein is correlated with a higher grade of glioma [84]. the silencing of HNRNPC inhibited proliferation and invasion, decreasing the expression of mi-R-21 and thus increasing the expression of its target programmed cell death 4 (PDCD4), a protein associated with tumour suppression [85]. A further study has corroborated this finding, with the knockdown of HNRNP A2/B1 reducing glioblastoma cell viability, invasiveness, and chemoresistance for temozolomide [84]. Its knockdown also induced apoptosis and reactive-oxygen-species generation in glioma cell lines, with a putative mechanism in the reduced expression of phospho-STAT3 and MMP-2. Protein–protein interaction analysis utilising immune-precipitation and mass spectroscopy has also demonstrated the association of HNRNPC and HNRNPA2/B1 with SOX2 in glioblastoma cell lines [86].

The insulin growth factor 2 mRNA binding proteins are another family of indirect m6A readers with implications in glioma pathology. IGF2BP1 has been shown to be upregulated in glioma and is associated with increased proliferation and invasiveness. Multiple long non-coding and micro RNAs targeting IGF2BP1, 2 and 3 have been implicated in glioma, using both patient-derived cell lines and xenograft models [87,88,89,90,91,92]. IGF2BP2 has been shown to regulate oxidative phosphorylation in glioblastoma, with shRNA silencing of IGFBP2 reducing the oxygen consumption rate and inhibiting tumour sphere formation [93]. A further study demonstrated that IGF2BP2 stabilises CASC9 to accelerate aerobic glycolysis in glioblastoma [94]. YTHDF2 may have a role in glioblastoma adaptation to the tumour microenvironment, facilitating the maintenance of GSCs in the hypoxic niche [92] and contributing to temozolomide resistance [90,95].

## 3. Therapeutic Targets

The role of m6A modification in promoting tumorigenesis across many cancer types has established it as an emerging novel therapeutic target [96,97].

### 3.1. FTO Inhibitors

An existing FDA-approved FTO inhibitor, the NSAID meclofenamic acid (MA2), is readily available and contributed to the initial studies investigating m6A regulation as a therapeutic target. In vitro studies in glioblastoma cell lines, and in vivo studies using a mouse model, have shown that the administration of MA2 suppresses glioblastoma stem-cell growth and proliferation [58]. A pilot clinical study investigating the effect of MA2 on patients with progressive or recurrent brain metastasis is currently underway and is due to be completed in April 2023 (NCT02429570). Alongside the readily available MA2, virtual screening and biochemical analyses have also been used to identify novel FTO inhibitors [98]. An example is Rhein, which was found to competitively bind to the catalytic subunit of FTO thus inhibiting downstream demethylation and leading to an overall significant increase in m6A levels [99]. Another novel FTO inhibitor is CHTB, a molecule which competitively inhibits FTO demethylase activity in a dose-dependent manner. In glioblastoma, a recent study demonstrated the efficacy of another FTO inhibitor, FTO-04, at impairing the proliferation of GSCs whilst preserving neural stem cells [100,101].

Alongside FTO, the m6A eraser ALKBH5 has also been identified as a therapeutic target. However, the trial of an ALKBH5 inhibitor in the glioblastoma cell line A-172 demonstrated negligible effects [102]. In contrast, another group used imidazobenzoxazin-5-thione MV1035, synthesized as a new sodium channel blocker but with inhibitory effects on ALKBH5 demonstrated by in silico and in vitro experiments [103]. This group demonstrated that the inhibitor reduced migration and invasiveness in the U87 glioblastoma cell line. A more recent study has corroborated the efficacy of such an approach, demonstrating two novel inhibitors of ALKBH5, Ena15 and Ena21, that inhibited cell proliferation of glioblastoma cell lines [104]. They further demonstrated that such inhibition increased m6A levels and stabilised the FOXM1 mRNA transcript, suggesting a putative mechanism of action.

### 3.2. METTL3 Inhibitors

There is extensive evidence implicating a role for METTL3 in the pathogenesis of GBM. However, research into the consequences of m6A regulation in GBM have demonstrated contrasting findings with regard to the tumorigenic or suppressive effect of METTL3. Indeed, both METTL3 inhibitors and activators have been suggested as potential therapeutic approaches. A small molecule which activates RNA m6a methylation by cooperatively binding to the methyltransferase writer complex has been described [105]. On the other hand, METTL3 inhibition has attracted considerable interest in the ongoing identification of novel compounds [106,107,108]. In a landmark for the translation of m6A biology into the clinic, a trial of a METTL3 inhibitor is ongoing in a cohort of acute myeloid leukaemia patients [109]. This compound has demonstrated in vivo and in vitro efficacy in suppressing the growth of AML, raising the prospect of utilising such approaches in other cancers.

The pharmacological modulation of m6A readers have also demonstrated promise in glioma. The drug linsitinib is an IGF1/IGF1R inhibitor which was demonstrated to preferentially target YTHDF2-expressing cells, inhibiting GSC viability and impairing glioblastoma growth in vivo [77]. IGFBP3 was identified as an indirect target of YTHDF2 via MYC, with YTHDF2 depletion causing the downregulation of IGFBP3 mRNA and protein levels. The compound B-asarone has been identified as an inhibitor of HNRNPA2/B1 and was able to supress glioma by inhibiting invasion and inducing cell-cycle arrest and apoptosis [110]. Overcoming the blood–brain barrier has long been a challenge for systemic treatments targeting gliomas, and intriguingly, the silencing of IGF2BP2 was demonstrated to increase blood–brain barrier permeability and enhance the antitumour efficacy of doxorubicin [111].

### 3.3. Inducible Editing of m6A Modifications via CRISPR-Cas Strategies

Whilst most drug research has targeted the proteins broadly regulating m6A modification, the inducible editing of m6A offers the prospect of targeted manipulation in a site-specific manner, avoiding unwanted off-target effects [112]. Such an approach recognises that the distribution of m6A on mRNA is variable and confers differential functions, depending on the specific transcripts and readers implicated. Programmable RNA m6A editing has been accomplished utilising CRISPR-Cas9 fused with a single-chain m6A methyltransferase [113]. Programmable site-specific removal of m6A was also accomplished via fusing CRISPR-Cas9 with ALKBH5 or FTO [113,114]. Other groups have adapted this approach using Cas13 fused to METTL3, FTO and ALBKBH5 [115,116,117,118]. Thus, engineered m6A writers and erasers enable the efficient manipulation of individual m6A sites in specific genes, which will both help further our understanding of their biological functions and potentially offer a novel treatment approach [119,120]. Recent work has developed light-inducible and ABA chemically induced-proximity m6A editing techniques, which permit an even finer degree of spatiotemporal control in m6A editing [112,121].

## 4. R-Loops

Genomic instability, an increase in the rate of genomic alterations, is a key mechanism implicated in the initiation and progression of many types of tumours, including GBM [122,123,124]. Genomic alterations which promote tumour development can facilitate the loss of critical tumour suppressor genes, cell-cycle checkpoint genes or the activation of oncogenes, and can arise as a result of unrepaired or incorrectly repaired DNA damage [125,126]. Genomic instability arises from diverse, distinct processes, including nucleotide instability, microsatellite instability and chromosome instability [127,128]. Sources of genomic instability include mitochondrial stress, DNA replication stress and telomere shortening [129,130,131,132]. R loops are transient and reversible tripartite structures comprising a DNA/RNA hybrid and a single strand of displaced DNA, which have recently also been implicated as a key source of genomic instability [133,134,135]. Importantly, it has recently been shown that the RNA strand in R loops can be m6A-methylated, and thus this mechanism may play a role in the pathogenesis of GBM (Figure 2) [136].

The formation of R loops is co-transcriptional, facilitated by the negative supercoiling that occurs behind RNA polymerase II [137,138]. R loops span 100–2000 base pairs and are prevalent at centromeres, telomeres, retrotransposons and the promoters and terminators of highly transcribed genes, including known oncogenes [133,139,140,141]. R loops are important for an array of normal cellular processes including transcription initiation and termination, DNA replication, the maintenance of telomeres and immunoglobulin class switching. Although the formation of R loops is essential for the regulation of gene expression and other cellular processes, their persistence in the genome is known to cause DNA damage, including double-strand breaks (DSBs) when R loops are cleaved by nucleotide excision repair proteins, by causing transcription-replication collision [142,143,144,145]. As such, the levels of R loops in cells must be tightly regulated to prevent accumulation, which could otherwise contribute to genomic instability [146]. To maintain the intracellular levels of R loops and prevent their accumulation, cells utilise both RNase H1 and RNase H2, which act to degrade the RNA strand in R loops via 5′-3′ exonuclease activity, and numerous helicases which unwind R loops, such as DExD-Box helicase 9 (DHX9), senataxin (SETX), the ATP-dependent helicase PIF1 and Aquarius [138,147,148].

The methylation of the RNA strand in R loops was demonstrated utilising immunoprecipitation techniques including m6A DIP and S9.6 DRIP, revealing that under normal physiological conditions, R loops containing m6A residues exist in human pluripotent stem cells [149]. The enrichment of R loops was discovered to arise during the S and G2/M phases of the cell cycle and the levels of R loops were found to be depleted during the G0/G1 phase. In LINE-1 and intronic regions of dividing cells, the m6A reader YTHDF2 was found to bind to m6A-methylated R loops to facilitate their removal from cells [149,150]. The knockdown of YTHDF2 and the m6A writer METTL3 was revealed to increase the levels of R loops and the DSB marker γH2ax, thus signifying that both proteins are crucial for degrading m6A-containing R loops and preventing DNA damage and genomic instability [140,149,151].

It has been shown that YTHDF2 is upregulated in GSCs and GBM tumours [77]. The accumulation of R loops is linked to the pathogenesis of cancer, and thus the dysfunction of YTHDF2 may act via the prevention of R loop removal to contribute to genomic instability and GBM tumorigenesis [146]. In contrast to the findings by Abakir et al. [149], others have argued that m6A methylation promotes the formation of R loops under normal physiological conditions and facilitates the efficient termination of transcription in HeLa cells [135,140]. The m6A writers METTL3 and WTAP were found to be upregulated in GBM and promote tumour growth and progression, and it may be that their upregulation contributes to increased m6A-containing R loops, and subsequent genomic instability, in GBM [54,152].

The presence of DNA damage in cells, such as DSBs, is known to contribute to a variety of diseases including cancer [153,154]. In the context of DNA damage, Xiang et al. discovered that in response to UV irradiation, m6A methylation occurs at sites of single-stranded DNA damage in order to facilitate the recruitment of DNA polymerase κ (Pol κ) for DNA repair [155,156]. R loops form at sites of DSBs, thus corroborating the notion that the presence of DNA damage can also induce the formation of R loops [156,157]. More specifically, they found that METTL3 becomes phosphorylated by the ataxia telangiectasia mutated (ATM) kinase at serine 43 when DSBs are present, thereby facilitating the m6A methylation of nascent RNA at sites of DNA damage. The m6A reader YTHDC1 is recruited and R loops form which enable the recruitment of BRCA1 and RAD51 to permit the HR-mediated repair of DSBs and prevent genomic instability [156,158]. Zhu et al. recently found that the knockdown of YTHDC1 increases the expression of VPS25 and promotes the proliferation of glioma cells, and therefore the METTL3-m6A-YTHDC1 axis could also be impaired in GBM [81]. As YTHDC1 is known to be differentially expressed in GBM, YTHDC1 expression may be significantly reduced and so contribute to genomic instability by preventing the repair of DNA damage [159].

As an alternative mechanism, Kang et al. recently revealed that the TonEBP-dependent m6A methylation pathway predominates at sites of DNA damage caused by UV irradiation or camptothecin [160]. In this pathway, the tonicity-responsive enhance binding protein (TonEBP), which is a regulator of transcription and a stress protein, is recruited to sites of DNA damage [161]. TonEBP binds via its Rel homology domain (RHD) to METTL3 and recruits this protein to R loops for m6A methylation. Once this has taken place, R loops are resolved via the recruitment of the endonuclease RNase H1 [160,162]. Thus, this molecular mechanism appears to arise in the presence of DNA damage to prevent the excessive accumulation of R loops and genomic instability. It is interesting to speculate that the TonEBP-dependent m6A methylation pathway could also potentially be impaired in GBM. Cui et al. discovered that the knockdown of METTL3 promotes the growth, self-renewal and tumorigenesis of GSCs, and as this protein is differentially expressed, the silencing of this m6A writer could potentially impair the pathway and thereby promote tumorigenesis in GBM [58,159].

Although m6A methylation has been revealed to be implicated in the formation and resolution of R loops under normal physiological conditions and in the presence of DNA damage, it remains unknown whether this process contributes to the formation and accelerated accumulation of R loops or to their resolution in GBM. However, it is interesting to note that several researchers such as Zhang et al. and Stork et al. have found that R loops can accumulate in cancer cells and promote DNA damage, genomic instability and tumorigenesis [157,163,164]. In contrast, other researchers such as Ye et al. and Prendergast et al. have found that the resolution of R loops can promote tumorigenesis by preventing DNA damage, genomic instability and the apoptosis of cancer cells [161,165]. Therefore, R-loop formation and resolution appears to be context-dependent and thus likely depends on a multitude of factors such as tumour location, tumour type, the expression levels of target genes and the expression levels of m6A writers, readers and erasers [166]. R-loop stability merits further focus in the study of GBM biology, with the potential to aid the development of novel therapeutics for these devastating tumours.

## 5. Future Directions

The methylation of RNA is a ubiquitous biological process which is perturbed in a range of cancers and, as such, emerging therapeutic strategies in these cancers may be of relevance to the treatment of glioblastoma [167]. In particular, the immune system forms a crucial component of the tumour microenvironment and the role of m6A methylation in the regulation and maintenance of tumoral immune cells is an important avenue for further research. Whilst modern immunotherapy agents have demonstrated great promise in the treatment of lung and melanoma cancers, patient response is highly variable in glioblastoma [168]. Complex immune escape mechanisms contribute to this variable response and may, in part, be mediated by m6A-regulated processes. For example, MYC, a direct regulator of the key immune checkpoints CD47 and PD-L1, is itself regulated by the effect of m6A on the C-myc pathway [169]. METTL3, FTO, and IGF2BP2 have been shown to function as tumour promoters through the C-myc pathway in an m6A-dependent manner in a range of cancers [169]. Furthermore, PDL-1 and PD-1 expression is correlated with the expression of multiple m6A writers, readers and erasers [169]. In a similar vein, Wnt/β-catenin signalling is implicated in the development of chemoresistance and has been shown to be modulated by m6A modification via the upregulation of TRIM11 in nasopharyngeal carcinoma [170].

The combination of m6A-targetting drugs such as FTO- and METTL3-inhibitors alongside existing anti-cancer therapy has been shown to sensitise tumours to treatment, demonstrated in animal models of a range of cancers types [171,172,173]. Such combinatorial therapies are an attractive potential strategy for novel glioblastoma treatments, and further research must better characterise the role of m6A within the privileged CNS immune microenvironment and assess the efficacy of synergistic therapies in glioblastoma animal models. METTL3 has been shown to maintain temozolomide resistance in glioblastoma cell lines and rodent models, suggesting the feasibility of using METTL3-inhibitors alongside current chemotherapy [57].

Whilst most translational research has focused on developing specific inhibitors against m6A regulatory proteins, exciting recent research has demonstrated the feasibility of packaging dCasRx–m6A editors within a lentiviral/AAV vector to robustly alter m6A methylation at multiple sites on target mRNA transcripts, demonstrating tumour suppression in a mouse model of bladder cancer [174]. Future research in glioblastoma must identify biologically relevant candidate target transcripts, against which such gene therapies can be designed.

## 6. Conclusions

There is extensive evidence implicating a role for the m6a RNA modification and its associated regulators in GBM, whether that be in driving or suppressing tumour progression. Future research should be directed at further uncovering the specific mechanisms through which m6A regulators influence GBM progression and identifying the targeted mRNA transcripts. This will then inform the discovery of small-molecule inhibitors and potentially identify targets for programmable m6A editing. The context-dependent effect of m6a modification will also pose a significant challenge when considering the potential clinical efficacy of targeting m6a regulators. Single-cell m6A sequencing has helped delineate the heterogeneity of this modification [4], and such an approach could be adapted for glioma. The downstream effect of m6A modification levels is complex, with both m6A writers and erasers implicated in glioma tumorigenesis. Recent work has described stoichiometric and positional analysis of m6A at single-nucleotide resolution across the whole transcriptome [175]. Alongside programmable editing, such approaches will be vital in delineating the mechanisms by which m6A modification contribute to tumour biology. Mass-spectrometry-based proteomics is increasingly utilised to define the protein-level effects of genetic mutations in cancer, and such approaches may delineate the downstream effects of post-transcriptional m6A mRNA modification on gene-protein expression [176,177].

## Figures and Tables

**Figure 1 cancers-15-03307-f001:**
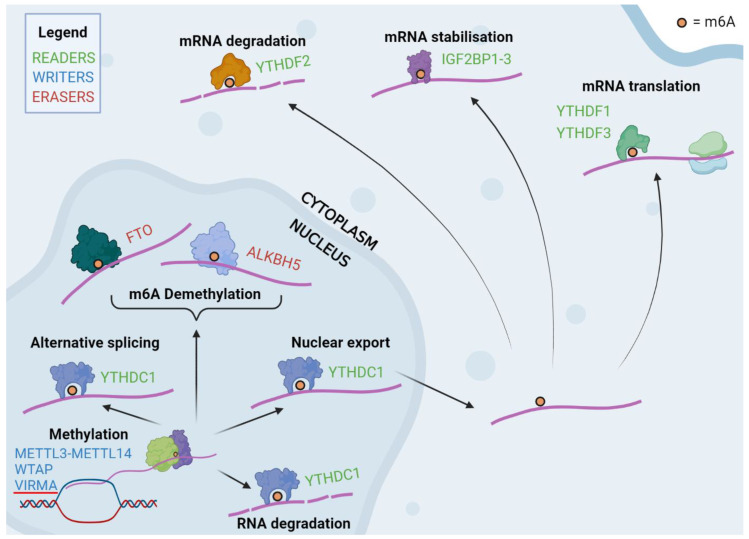
The METTL3–METTL14 heterodimer and associated proteins form the m6A writer complex, whereas FTO and ALKBH5 erase the modification. The m6A modification has a variable effect on mRNA fate, mediated via specific reader proteins.

**Figure 2 cancers-15-03307-f002:**
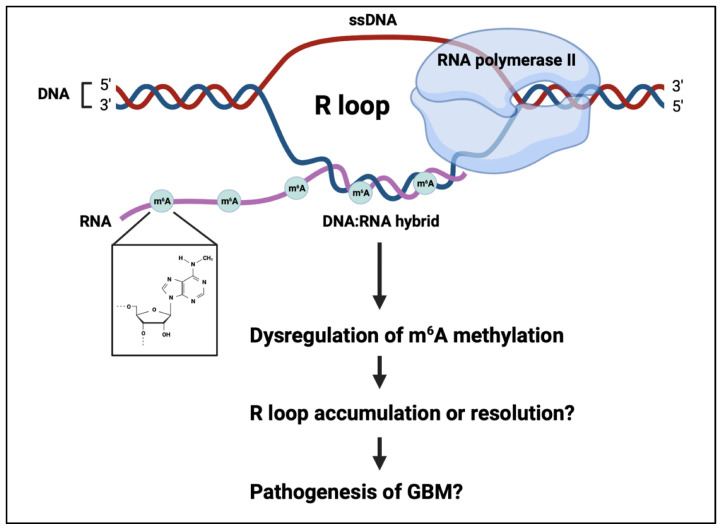
R-loops are a tripartite structure consisting of a DNA:RNA hybrid and a strand of displaced DNA. The RNA can be m6A methylated, modifying R-loop fate. This process may have a role in genomic instability in the pathogenesis of glioblastoma.

## Data Availability

Not applicable.

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
