# Peer review of "The Regulation of m6A Modification in Glioblastoma: Functional Mechanisms and Therapeutic Approaches"

_cancers, 2023, doi:10.3390/cancers15133307_

Round 1

Reviewer 1 Report

The manuscript is a review focused on the N6-methyladenosine (m6A) modification of mRNA in glioblastoma. The article is unique because m6A modification is centered on glioblastoma, whereas other reviews do not emphasize glioblastomas. Thus, this review fills a gap with a thorough review of glioma m6A alternations not covered in other reviews. Because the article is a review, there is no methodology employed that needs to be improve. Throughout the article, the authors presented findings from multi-labs with labs agreeing on the role of m6A in glioblastoma and included results in which labs presented different interpretations of m6A roles. The manuscript is well-referenced with the appropriate references. The figures are adequate. However, Figure 1 could be enhanced if the m6A-associated proteins could have a writer, eraser, or reader in parathesis next to their name in the figure.

Reviewer 2 Report

Cancers – MDPI – 10 June, 2023

The role of a disordered regulation of the N6-methyladenosine (m6A) RNA modification in glioblastoma and its consequences was revised in this manuscript. Participation of the proteins that methylate, demethylate, and recognize the m6A modification (writer, eraser, and reader categories, respectively) is emphasized and contrasting findings with regards to the tumorigenic or suppressive effect of METTL3 were well discussed. How these m6A regulators influence glioblastoma progression is a point the authors address as a realm to future research. The text is well written and the literature provided was plentiful.

Considering that RNA modification is altered in brain tumors and that a better understanding of tumor biology is urgently required, this review is welcome.

Minor:

Lines 72, 80: Standardize the reference style in the text (citation number)

References:

Article pages or electronic numbers are missing in citations 1, 17, 26, 36, 41, 43, 46, 50, 51, 53, 56, 58, 65, 72, 73,74, 75, 82, 93, 94, 102, 109, 111, 119, 12, 124, 125, 126, 128, 130, 142, 157, 159, 160, 162, 163

Highlight the article year in citations 5, 6, 8, 15, 17, 18, 23, 26, 27, 33, 38, 40, 42, 43, 58, 65, 66, 80, 87, 95, 96, 106, 122, 123, 124, 128, 132, 134, 135, 139, 140, 144, 145, 146, 147, 149, 155, 156, 157, 164

Incomplete citations: 44, 79, 81, 87, 113, 116, 137, 148, 152, 161

Citation 131: “DNA”

Reviewer 3 Report

The review “The Regulation of m6A modification in Glioblastoma: functional mechanisms and therapeutic approaches” by Simon Deacon is timely and well-written.

The authors discuss possible role of the most common post-transcriptional RNA modification in eukaryotic cell and a key regulator of RNA fate- the N6-adenosine methylation (m6A) – in glioblastoma pathogenesis. The review is based on hypothesis that given its major role in neural system homeostasis, aberrant m6A might be one of the drivers of the brain cancers (and the glioblastoma is a very common and aggressive type of brain cancer)

The authors provide a comprehensive overview of the enzymes involved in methylation, demethylation and recognition of the m6A mark, the specific aspects of their functioning and levels in normal cells and in glioblastoma (and glioblastoma “cancer stem cells”). They also discuss the role of m6A mark in the formation and resolution of R loops (RNA/DNA hybrid and a single strand of displaced DNA), including in case of glioblastoma.

Please find my suggestions below.

I suggest the authors propose several strategies to overcome possible challenges when developing therapeutics targeting m6A RNA modification pathways. This will demonstrate the feasibility of the future translational studies in this field.

Line 224. “has shown hypoxia-induced upregulation of ALKBH5 in a HIF-1α dependant”. In this sentence the HIF-1α is in italic, as if it was the name of the gene, not the protein. Please clarify.

Also, the authors review studies both in humans and in laboratory animals. It would be beneficial for the readers if the authors clarified in the text whether they are talking about human cells/enzymes etc, of about mice/etc, at least when discussing the key findings.

Line 254. “levels of m6a RNA” please fix the typo (the correct abbreviation is m6A)

When talking about proteins/microRNAs/other biomolecules potentially involved in regulation of m6A and hence potentially in glioblastoma pathogenesis, the authors might want to suggest pharmacological approach(es) to modulation of such regulators or make a statement that there are currently no such pharmacological modulators.

Line 423. The authors state that “R loops <>
are prevalent at centromeres, telomeres, retrotransposons and the promoters and termi
nators of highly transcribed genes”.
They also might want to mention that many of such genes are known oncogenes (for example, cMyc). What about glioblastoma driving oncogenes? Do they also have sequences prone to R loop formation?

Can the “lessons from other cancers” with similar pathogenesis be extrapolated to glioblastoma, in terms of the role of RNA m6A mark? The authors referred to some publications about breast cancer though.

Perhaps, adding the brief Future directions section will straighten the text.
